# Soft Tissue and Biomolecular Preservation in Vertebrate Fossils from Glauconitic, Shallow Marine Sediments of the Hornerstown Formation, Edelman Fossil Park, New Jersey

**DOI:** 10.3390/biology11081161

**Published:** 2022-08-02

**Authors:** Kristyn K. Voegele, Zachary M. Boles, Paul V. Ullmann, Elena R. Schroeter, Wenxia Zheng, Kenneth J. Lacovara

**Affiliations:** 1Department of Geology, Rowan University, Glassboro, NJ 08028, USA; bolesz@rowan.edu (Z.M.B.); ullmann@rowan.edu (P.V.U.); lacovara@rowan.edu (K.J.L.); 2Jean and Ric Edelman Fossil Park, Rowan University, Mantua Township, NJ 08080, USA; 3Department of Biological Sciences, North Carolina State University, Raleigh, NC 27695, USA; easchroe@ncsu.edu (E.R.S.); wzheng2@ncsu.edu (W.Z.)

**Keywords:** soft tissues, molecular preservation, collagen, Hornerstown Formation, shallow marine, glauconite

## Abstract

**Simple Summary:**

Original organics and soft tissues are known to persist in the fossil record. To date, these discoveries derive from a limited number of ancient environments, (e.g., rivers, floodplains), and fossils from rarer environments remain largely unexplored. We studied Cretaceous–Paleogene fossils from a peculiar marine environment (glauconitic greensand) from Jean and Ric Edelman Fossil Park in Mantua Township, NJ. Twelve samples were demineralized in acid to remove the mineral component of bone. This treatment frequently yielded products that are visually consistent with bone cells, blood vessels, and bone matrix from modern animals. Fossil specimens that are dark in color exhibit excellent microscopic bone preservation and yielded a greater recovery of original soft tissues, whereas light-colored specimens exhibit poor microscopic preservation and yielded few to no soft tissues. Additionally, a well-preserved femur of a marine crocodilian was found to retain original bone protein by reactions with antibodies. Our results: (1) corroborate previous findings that original soft tissue and proteins can be recovered from fossils preserved in marine environments, and (2) expand the range of ancient environments documented to preserve original organics and soft tissues. This broadens the suite of fossils that may be fruitful to examine in future paleomolecular studies.

**Abstract:**

Endogenous biomolecules and soft tissues are known to persist in the fossil record. To date, these discoveries derive from a limited number of preservational environments, (e.g., fluvial channels and floodplains), and fossils from less common depositional environments have been largely unexplored. We conducted paleomolecular analyses of shallow marine vertebrate fossils from the Cretaceous–Paleogene Hornerstown Formation, an 80–90% glauconitic greensand from Jean and Ric Edelman Fossil Park in Mantua Township, NJ. Twelve samples were demineralized and found to yield products morphologically consistent with vertebrate osteocytes, blood vessels, and bone matrix. Specimens from these deposits that are dark in color exhibit excellent histological preservation and yielded a greater recovery of cells and soft tissues, whereas lighter-colored specimens exhibit poor histology and few to no cells/soft tissues. Additionally, a well-preserved femur of the marine crocodilian *Thoracosaurus* was found to have retained endogenous collagen I by immunofluorescence and enzyme-linked immunosorbent assays. Our results thus not only corroborate previous findings that soft tissue and biomolecular recovery from fossils preserved in marine environments are possible but also expand the range of depositional environments documented to preserve endogenous biomolecules, thus broadening the suite of geologic strata that may be fruitful to examine in future paleomolecular studies.

## 1. Introduction

Numerous molecular paleontological investigations have been conducted on geologically ancient fossils, (i.e., >1 Ma years old; [1]), many of which have demonstrated strong evidence for the preservation of endogenous biomolecules and soft tissues in deep time. These studies have yielded the recovery of cellular and tissue structures morphologically similar to their extant counterparts, (e.g., [2,3,4,5,6,7,8,9,10,11,12,13,14]), identified the presence of proteins in fossil soft cells and soft tissues via a variety of techniques, (e.g., [4,5,6,10,15,16,17,18,19,20]), and even recovered protein sequences from mass spectrometry, (e.g., [4,21,22,23,24]). The wealth of such discoveries, and the number of fossil taxa and preservational environments investigated and found to yield soft tissue and biomolecular preservation continues to grow. Initially, only terrestrial environments were investigated, as it was thought that hydrolysis from continuous exposure to water would be inconducive to molecular preservation [25,26]. More recently, a few studies examined aquatic taxa or fossil specimens preserved in marine environments, (e.g., [6,16,27]). However, the number of geologic formations and types of host lithologies, and, therefore, the types of depositional paleoenvironments investigated, still remain limited. 

To date, the majority of marine specimens that have been analyzed derive from siliciclastic deposits of various geologic ages (Triassic to Neogene), degree of consolidation (unconsolidated vs. consolidated), and grain size (see Table 1). A few samples have also been examined from shallow marine limestones [16] and chalk [6]. Wiemann et al. [10] also analyzed an indeterminate crocodilian specimen from glauconite containing sands of the Cretaceous Navesink Formation. Only this crocodilian, one of the *Nothosaurus* specimens examined by Surmik et al. [16], and the *Ichthyosaurus* specimen examined by Wiemann et al. [10] were reported to not yield preserved soft tissues. Surmik et al. [16] and Lindgren et al. [6] investigated a suite of specimens from marine environments further for the identification of proteins and found organic compounds, including fragments of amino acids, and endogenous collagen I, respectively. These studies demonstrate that marine fossils from various environments can also preserve soft tissues and biomolecules. 

Herein, we examine marine fossils collected from the Jean and Ric Edelman Fossil Park (EFP) at Rowan University in Mantua Township, New Jersey. All specimens were recovered from the Late Cretaceous (Maastrichtian)–early Paleogene (Danian) Hornerstown Formation. All but two of the examined specimens derive from the same assemblage: the Main Fossiliferous Layer (MFL). The MFL is a rich, 10 cm thick, multitaxic bonebed that begins approximately 20 cm above the base of the Formation [28]. As in the prior paleomolecular studies of marine bones discussed above, the Hornerstown Formation is a siliciclastic deposit that was laid down on an organic-rich, shallow-marine shelf [29]. Specifically, the Hornerstown Formation is comprised entirely of heavily bioturbated, unconsolidated, glauconitic greensands [29,30]. It contains comparatively more of this iron- and phosphate-rich, diagenetic mineral, glauconite, than the underlying Navesink Formation, from which a crocodilian specimen was investigated by Wiemann et al. [10]. The wealth of vertebrate fossils in the MFL within the Hornerstown Formation [29,31] allowed us to further investigate the possibility of soft tissue and biomolecular preservation in glauconite-forming shallow marine environments. Additionally, the multitaxic character of the MFL assemblage permitted analysis of specimens from several clades [28], thus allowing depositional and diagenetic effects to be controlled for when making comparisons among taxa. Voegele et al. [28] also demonstrated that the diversity and abundance of fossils within the MFL, as well as their stratigraphic distribution and taphonomic attributes, indicate that the MFL likely represents a mass-death assemblage, meaning that time averaging among specimens is minimal (also see [30]).

We considered the Hornerstown Formation as a lithosome potentially favorable for the preservation of endogenous organics because glauconite is rich in iron, which has been suggested to aid in molecular preservation. Specifically, it has been hypothesized that dissolved iron in diagenetic pore fluids reacts with peroxides sourced from decaying lipids to form iron free radicals, which in turn can induce chemical chain reactions resulting in crosslinking of biomolecules, their decay products, metal cations in solution, and dissolved humics [3,32,33]. Therefore, we examined twelve specimens of five taxa from the MFL and higher within the Formation for soft tissue and biomolecular preservation. This included analyzing one specimen of the marine crocodile *Thoracosaurus* for the preservation of the primary structural protein collagen I using molecular assays.

**Table 1 biology-11-01161-t001:** Summary of fossil specimens collected from marine sediments that have previously been examined for cellular and/or soft tissue preservation by demineralization.

Citation	Specimen Number	Taxon	Age	Formation	Rock Type
Barker et al. [34]	PAS11-02	Cetacea indet.	Miocene	Pungo River	Unconsolidated Silt and Clay
Barker et al. [34]	PAS11-03	Cetacea indet.	Miocene	Pungo River	Unconsolidated Silt and Clay
Barker et al. [34]	PAS11-01	Cetacea indet.	Miocene	Calvert	Unconsolidated Silt and Clay
Barker et al. [34]	PAS11-04	Dugongidae indet.	Miocene	Torreya	Unconsolidated Silt and Clay
Cadena and Schweitzer [35]	CCMFC01	Pelomedusoides indet.	Paleogene	Los Cuervos	Sandstone
Lindgren et al. [6]	IRSNB 1624	*Prognathodon*	Cretaceous	Ciply	Phosphatic Chalk
Macauley et al. [36]	RU-CAL-00001	Testudines indet.	Miocene	Calvert	Unconsolidated Silt and Clay
Schweitzer et al. [3]	WCBa-20	*Balaenoptera*	Miocene	Pisco	Mudstone
Surmik et al. [16]	WNoZ/s/7/166	*Nothosaurus*	Triassic	Gogolin	Limestone
Surmik et al. [16]	SUT-MG/F/Tvert/15	*Nothosaurus*	Triassic	Gogolin	Limestone
Surmik et al. [16]	SUT-MG/F/Tvert/2	*Protanystropheus*	Triassic	Gogolin	Limestone
Wiemann et al. [10]	YPM 656	Crocodylia indet.	Cretaceous	Navesink	Unconsolidated Greensand
Wiemann et al. [10]	YMP VP 059362	*Ichthyosaurus*	Jurassic	Posidonia Shale	Mudstone

## 2. Materials and Methods

### 2.1. Materials

Fossil specimens examined herein were collected from the Hornerstown Formation between 2011 and 2019; most were collected as part of a systematic grid excavation of the MFL [37]. In the field, all but one sample (*Thoracosaurus* scute RU-EFP-00006-8, drawn from collections) were set aside immediately after discovery for molecular analyses by wrapping each in sterilized aluminum foil and placing them in autoclave-sterilized mason jars with silica gel desiccant beads. Collection was performed while wearing nitrile gloves to limit contamination, and specimens were stored in climate-controlled buildings, thus minimizing temperature and humidity fluxes after collection. Sediment that was adjacent to, but not in contact with, each fossil was collected for use as a negative control and stored as above. To avoid the possibility that glue residues could be mistaken for biological structures, no glues or stabilizing agents were used during either excavation or preparation of specimens. Eight specimens were recovered from the MFL and two were recovered from the upper Hornerstown Formation. Taxonomically, the specimens derive from the turtles *Euclastes*, *Taphrosyphs*, and two other unidentified turtles, the gavialoid crocodilian *Thoracosaurus*, and another unidentified crocodilian. Further details about each specimen are provided in Table 2.

Specimen RU-EFP-00006 is a partially articulated skeleton of *Thoracosaurus neocesariensis*. The right femur (Figure 1A) was chosen to analyze for collagen I remnants because this bone is generally well-preserved in terms of gross morphology (Appendix A). Additionally, being a large limb bone, it possesses a greater amount of cortical tissue for analyses. These attributes imply this fossil could be a favorable candidate for molecular retention [25,38,39]. Stylopodial and zeugopodial elements of a juvenile American alligator (*Alligator mississippiensis*) served as a modern positive control for biomolecular assays (see Supp for treatment of extant bones prior to analyses below). Both modern and fossil molecular analyses were conducted with solely cortical tissue.

### 2.2. Methods

All fossil analyses were performed in a permanently dedicated, fossil-only laboratory at North Carolina State University (NCSU; see Schroeter et al. [24] for additional details on lab sterilization protocols). For the biomolecular analyses, modern control trials were conducted in a separate lab at NCSU. Over the duration of this project, demineralization was carried out by separate project personnel in three separate labs: one sample (*Thoracosaurus* femur RU-EFP-00006-11) was demineralized at NCSU, seven samples were demineralized in a designated, sterilized, fossil-only chemical fume hood at Drexel University, and four were demineralized in a similarly dedicated, sterilized chemical fume hood at Rowan University. Negative controls, which included buffer-only solutions and sediments demineralized and/or co-extracted in the same reagents, were evaluated in tandem with the fossil for all assays. For immunoassays, specificity controls were also conducted (see below). Procedures for all controls and modern/fossil samples were performed in exactly the same manner unless otherwise noted. Three replicates were completed of each assay.

Collagen I, a structural protein abundant in bone tissue, was selected as the primary target for biomolecular investigation in this study because of its high preservation potential. Many factors contribute to this high preservation potential, including: (1) that it is the most abundant protein in bone [40,41,42,43]; (2) its primary, secondary, and tertiary structures make it highly stable [25,44,45] and resistant to many proteases [46], and (3) its intimate association with hydroxyapatite crystallites within the tissue structure of bone, (e.g., [25,40,41,47,48]).

#### 2.2.1. Histology

Standard petrographic thin sections were prepared for histological analysis (see [49]). Histological analysis was performed on eight of the twelve specimens that were demineralized (three crocodilian and five turtle samples; see Table 2 for specimen specifics) to acquire a representative dataset of histological variability among specimens from EFP. Bone fragments were embedded in Silmar 41 resin (U.S. Composites), thick sectioned, and mounted to frosted glass slides with Loctite^®^ Heavy Duty 60 Minute epoxy (Westlake, OH, USA). Thin sections were then ground to an appropriate thickness (~100 µm) and polished. The Histological Index (HI) of Hedges and Millard [42] was used to qualify the degree of microstructural preservation.

#### 2.2.2. X-ray Diffraction (XRD)

XRD analyses were conducted using a Phillips X’Pert diffractometer (#DY1738) at the Department of Earth and Environmental Science at the University of Pennsylvania. Approximately 1–2 g of two fossil bone samples (one light tan, one dark purple-brown) and Hornerstown Formation sediment were each mechanically powderized to less than 10 μm in a SPEX tungsten carbide Mixer-Mills (model #8000). Each powdered sample was then loaded into a sample holder and analyzed using Cu Kα radiation (λ = 1.54178A°) and operating at 45 kV and 40 mA. Diffraction patterns were measured from 5–75° 2θ with a step size of 0.017° 2θ and 1.3 s per step (=0.77 degrees per minute). Phillips proprietary software HighScore Plus v. 3.0e was used to interpret the resulting diffraction patterns. 

#### 2.2.3. Demineralization

A roughly 0.5 cm^3^ fragment of each fossil was initially submerged in 0.5 M disodium ethylenediaminetetraacetic acid (EDTA) to chelate calcium. This solution was changed every other day for approximately four weeks, then weekly for another 1–2 months, as needed. In some cases, demineralization of fossils from this locality was found to occur at a drastically slower rate than we have previously encountered for other similarly aged specimens, including others from marine sediments [11,34,36]. After 1–2 months in EDTA, if demineralization was still mostly ineffective, then as necessary samples were demineralized by acid dissolution in 0.2 M or 0.5 M hydrochloric acid (HCl) for several days to two weeks, with solution changes made every 1–2 days, depending on the resilience of the fossil bone matrix. Demineralization products were transferred to glass slides and visualized using a Zeiss Stemi 2000-C reflected light microscope (Oberkochen, Germany) with a linked AxioCam 506 camera (Zeiss, Oberkochen, Germany). Abundances of osteocytes, vessels, and fibrous matrix fragments from each specimen were assigned to five relative categories based on Ullmann et al. [11]: absent, rare, uncommon, frequent, and abundant. Modern alligator and Hornerstown Formation sediment samples were also demineralized following the same procedures, to serve as positive and negative controls, respectively.

#### 2.2.4. Scanning Electron Microscopy (SEM) and Energy Dispersive X-ray Spectroscopy (EDX)

Osteocytes isolated by demineralization in 0.2 M HCl from RU-EFP-00006-11 were collected using a 1 μm filter (Millipore), which was then placed on an aluminum stub and allowed to dry. The resulting uncoated sample was imaged using a field emission-scanning electron microscope (FE-SEM Zeiss Supra 50VP, Oberkochen, Germany) at Drexel University, operating at a working distance of 6.6 mm and at an accelerating voltage of 1 kV. Elemental spot analyses were collected as a standardless assay using a coupled Oxford model 7430 EDS INCAx microanalyzer (Oxford Instruments, Abingdon, UK).

#### 2.2.5. Protein Extraction

Herein, we used the same protein extraction protocol as Ullmann et al. [20]. In brief, ground bone and sediments, along with a buffer/“blank” control of only the extraction solutions, were each separately extracted with 0.6 M HCl. The resulting supernatants were collected (here called the “HCl fraction”), then the remaining pellet was incubated in 0.05 M ammonium bicarbonate (AMBIC). The resulting supernatants were then also collected (here called the “AMBIC fraction”). HCl fractions were precipitated with trichloroacetic acid (TCA) to form pellets, which were then allowed to dry in a laminar flow hood. AMBIC fractions were dried in a speed vacuum. Samples were stored at either 4 °C, −20 °C, or −80 °C depending on the length of time until analysis. Concentrations are reported based on amounts of pre-extracted bone rather than absolute masses because post-extraction yields from this protocol contain salts from extraction buffers in addition to varying amounts of protein [50]. Fresh *Alligator* cortical bone was extracted in the same manner as the fossil using the same reagents and the same protocol, but in a separate lab at NCSU.

#### 2.2.6. Polyacrylamide Gel Electrophoresis (PAGE) with Silver-Staining 

To test for the presence of organics within the extracts from *Thoracosaurus* femur RU-EFP-00006-11, we followed the silver-staining protocols of Zheng and Schweitzer [51], as also detailed in Ullmann et al. [20], but without treatment with the iron chelator pyridoxal isonicotinic hydrazide as it was not found to be necessary (refer to the Appendix A for details). 

#### 2.2.7. Enzyme-Linked Immunosorbant Assay (ELISA)

Herein, we used the same ELISA protocol as Ullmann et al. [20]. This included plating of AMBIC extracts at a concentration equivalent to 200 mg of pre-extracted bone per well, blocking of non-specific binding, and incubation with rabbit anti-*Alligator* purified-skin collagen antibodies. Extracts from an extant *Alligator* were analyzed using the same protocols (in a separate lab at NCSU) as a modern control (see the Appendix A for further details). 

#### 2.2.8. Immunofluorescence

We followed the fossil tissue embedding and immunofluorescence procedures of Schweitzer et al. [4,15], which are detailed in Zheng and Schweitzer [51]. This included embedding of RU-EFP-00006-11 demineralization products in LR White^TM^ resin (Electron Microscopy Sciences, Hatfield, PA, USA), gathering of 200 nm sections with an ultramicrotome, and incubation with rabbit anti-chicken collagen I antibodies. Inhibition and collagenase digestion specificity controls, as well as a modern *Alligator* control, were also conducted (see the Appendix A for further details).

## 3. Results

### 3.1. Histology

Histological investigation of the eight sectioned specimens revealed variable preservation of original bone microstructure among and within samples. The gross morphology of all fossil specimens examined herein is well preserved. However, the external surface of nearly half of the specimens is light in color while that of the other half is dark (Table 2). Variation in color even occurs among bones of the same individuals, such as crocodile RU-EFP-00030 (Table 2). Areas of disrupted microstructure correspond to areas of bone that are light in color under gross morphological investigation, which comprise varying amounts of the total bone thickness/volume (see below). This light alteration is common in fossil bones from this locality and has been previously noted by others ([52]). In long bones, these areas of discoloration occur on the outer cortical layer, (e.g., *Thoracosaurus* femur RU-EFP-00006-11; Figure 2) or extend the entire way across the cortex, (e.g., crocodile tibia RU-EFP-00030-1). Tabular-shaped bones exhibit the same pattern of surficial discoloration and disrupted microstructure; as in long bones, these alteration effects may be restricted to near the cortical margin, (e.g., *Thoracosaurus* scute RU-EFP-00006-8) or pervasive through the entire cross-section of a specimen, (e.g., *Taphrosphys* peripheral RU-EFP-02175). Within these discolored regions, occasional Wedel tunnels are present and the original microstructure is completely or almost completely obliterated, (e.g., Figure 1B,D). In contrast, the darker-colored areas of fossil bones exhibit well-preserved microstructure, with primary and secondary osteons, lines of arrested growth, and osteocyte lacunae clearly visible, (e.g., Figure 1A,C,F).

The microstructure of RU-EFP-00006-11 is well preserved (HI = 5) except for its outermost 50–250 μm (Figure 1A and Figure 2). In this outer rim, the bone is light in color and the microstructure is completely obliterated (HI = 0). A few Wedl tunnels extend from this poorly preserved rim into the underlying, darker, better-preserved cortex (Appendix A). The rest of the 5 mm of total cortical thickness is dark in color and well-preserved histologically. The outer cortex is composed of lamellar-zonal bone with at least 17 lines of arrested growth present. The innermost cortex is composed of compact coarse-cancellous bone. Osteocyte lacunae with short branching canaliculi are visible throughout the dark regions of the cortex. Small, longitudinal vascular canals are also present and in some instances are partially infilled with authigenic gypsum and pyrite.

Microstructure preservation among specimens varied, with some being well preserved, (e.g., RU-EFP-00006-11) and others poorly preserved (HI = 1 or 2). Preservation at the microscopic level does not correlate with macroscopic preservation, (i.e., well-preserved gross morphology does not always equate to high histologic integrity) or cortical thickness of a specimen. Specimens RU-EFP-02245, RU-EFP-00006-8, and RU-EFP-00002-2 exhibit a thin rim of this lighter, histologically degraded bone. In these specimens, as in RU-EFP-00006-11, the remainder of the cortex is dark in color and retains high histologic integrity. In contrast, crocodile tibia RU-EFP-00030-1 and *Taphrosphys* peripheral RU-EFP-02175 are entirely light in color with poor microstructure throughout, including large areas of bone tissue that appear completely obliterated by microbial destructions (possibly microscopical focal destructions [MFDs]; Appendix A; [53]). These alterations are invisible to the naked eye; the gross morphology of RU-EFP-00030-1 is well-preserved with little evidence of bioerosion or damage to the periosteal surface to indicate its microstructure is destroyed. Histological analysis has yet to be conducted on crocodile humerus RU-EFP-00030-2, but thick sections of this bone (made for another study) exhibit an overall moderate level of discoloration between that of the light bone with poor microstructures and dark well-preserved bone seen in other specimens. This is in contrast to the costal prong of RU-EFP-00018, which is dark throughout even though it has a very thin external cortex and an expansive cancellous region.

### 3.2. X-ray Diffraction

The diffractogram of a sample of RU-EFP-00006-11 identifies it to be composed of fluorapatite (Figure 3A), indicating it has been minorly altered from its original hydroxyapatite composition. Based on these results, combined with the generally well-preserved gross morphology and histology of RU-EFP-00006-11 and its production of relatively abundant demineralization products, we hypothesized that this specimen could be a favorable candidate for biomolecular analyses. The diffractogram of a light-colored, indeterminate crocodilian bone fragment from the upper Hornerstown Formation reveals that altered bone is also composed of fluorapatite. Both bone samples were distinct from Hornerstown Formation sediments, which were identified as glauconite.

### 3.3. Demineralization 

A few bone fragments, (e.g., RU-EFP-00002-2) were demineralized within a month in EDTA, but most remained relatively hard and brittle after six weeks and required further treatment in HCl before they could be imaged. In general, light-colored bone fragments demineralized more quickly than darker bone fragments and would develop a comparatively more granular and crumbly texture early in the process of breaking down (Appendix A). Both demineralizations with EDTA and HCl yielded structures morphologically consistent with osteocytes, vessels, and fibrous matrix. Although further molecular testing is needed to confirm that these structures represent original cells/tissues, we hereafter refer to them as osteocytes and vessels for brevity. Our antibody assays identified the presence of endogenous collagen I in pieces of fibrous matrix (see below), supporting the endogenous nature of these tissue fragments. Osteocytes were observed fully embedded, partially isolated, or fully isolated from mineralized bone matrix. No sediment samples yielded any structures morphologically consistent with vertebrate cells or tissues; only, small, subrounded to angular glauconite and quartz grains were observed (Appendix A). Several forms of osteocytes were recovered from fossil bone samples, varying in both cell body shape and complexity of branching of filopodia (Figure 4I–L). Morphologically, these osteocytes correspond to the stellate and flattened oblate morphologies as defined by Cadena and Schweitzer [27]. All morphotypes retain filopodia, some of which possess tertiary, quaternary, or even more ramifications in better preserved specimens, as present in osteocytes isolated from modern bone (Figure 4A). Most recovered “osteocytes” were red-brown in color, (e.g., Figure 4I–L), but a few samples, (e.g., RU-EFP-02175, RU-EFP-04161-8) yielded primarily dark brown to black osteocytes (from both EDTA and HCl demineralization, e.g., Appendix A).

Following the qualitative categories of Ullmann et al. [11], seven of the twelve demineralized fossils produced Abundant osteocytes, and RU-EFP-00006-8 produced Frequent osteocytes. In the four remaining samples, three (RU-EFP-00030-2, RU-EFP-02175, and RU-EFP-02245) produced only rare osteocytes, and osteocytes were absent in RU-EFP-00030-1. In fact, RU-EFP-00030-1 did not yield any cellular or soft tissue microstructures (osteocytes, vessels, or fibrous matrix). Among all the specimens tested, only three (RU-EFP-00006-11, RU-EFP-04161-8, and RU-EFP-00002-2) yielded vessels, ranging from ~20–50 µm in diameter, and in all three cases their recovery was either frequent or rare. These same three specimens, as well as RU-EFP-00006-8, also yielded frequent to the rare fibrous matrix. Unlike many EFP samples, RU-EFP-00006-11 yielded all three types of microstructures. The fibrous matrix was more plentiful when this bone was demineralized with HCl, but both HCl and EDTA yielded only a few vessels. 

### 3.4. Scanning Electron Microscopy and Energy Dispersive X-ray Spectroscopy 

SEM of microfiltered RU-EFP-00006-11 demineralization products revealed three-dimensional osteocytes with rough, fibrous surface texture over each cell body and short, branching filopodia (Figure 3B). Broken bases of filopodia appeared to exhibit brittle fractures and solid cross sections (not hollow). Regarding surface texture, similar rough, crisscrossing grooves on the surface of isolated osteocytes have previously been attributed to either the impression of collagen fibrils or microbial degradation [8,11,54]. EDX spot analyses found these microstructures to be composed of oxygen, iron, and carbon (Figure 3C), similar to the results of many other EDX studies of fossil bone demineralization products, (e.g., [8,11,55]. Under SEM, osteocytes from RU-EFP-00006-11 are indistinguishable from those of extant reptiles, (e.g., [27,56]). 

### 3.5. Polyacrylamide Gel Electrophoresis with Silver-Staining 

After electrophoresis, the movement of the AMBIC fossil sample through the gel colored the lanes a light yellow-brown (Figure 5A) prior to the application of chemical staining. Incubation in fixing solution (50% methanol) reduced but did not eliminate this “pre-staining” coloration (see Appendix A). Despite this, a drastic increase in color intensity was readily apparent across the entire examined a range of molecular weights after silver-staining (Figure 5B), indicating the presence of organics within the fossil bone AMBIC extracts. “Pre-staining” coloration was decreased by resuspending the portion of the bone pellet that did not dissolve into the solution used for the fossil lane and adding it as a second sample in another lane. Resuspended bone pellet lanes exhibited little to no “pre-staining” and a clear increase in signal intensity after development with silver nitrate (Figure 5A,B), again indicative of the presence of organics within this sample. 

Extraction buffer blanks and Laemmli buffer controls exhibited no staining. Sediment controls exhibited weak “post-development” staining (after incubation with silver nitrate) only at the highest and lowest molecular weights examined (Figure 5). In a subset of replicates, sediment lanes exhibited a faint band near 50 kDa; no banding or increase in staining was discernible in any fossil or other control lane at this molecular weight, indicating it represents a type(s) of organics present only within the sediment controls. In separate gels, modern *Alligator* extracts imparted no “pre-staining” and yielded clear, distinct banding patterns with minimal smearing after development via silver staining (Figure 5C).

### 3.6. Enzyme-Linked Immunosorbent Assay 

In all replicates, a positive signal for collagen I was identified in fossil AMBIC extracts by absorbance readings well over twice the background, (e.g., [57,58]). At 240 min, AMBIC extracts of RU-EFP-00006-11 reached an absorbance of 0.855 (Figure 6). At this same time point, sediment and extraction blank controls exhibited negligible absorbance. In all time point readings, the signal from the fossil bone was at least an order of magnitude higher than in all the negative controls. RU-EFP-00006-11 AMBIC extracts exhibited a significantly reduced signal relative to modern *Alligator* AMBIC extracts, which reached saturation (2.85) at ~150 min (Appendix A). At this same time point in the best replicate, the *Thoracosaurus* AMBIC sample reached an absorbance of 0.53. 

### 3.7. Immunofluorescence

Fossil demineralization products reacted positively with polyclonal anti-chicken collagen I antibodies. Fluorescence was only observed in primary antibody-incubated tissue sections and was well above the negligible background fluorescence of secondary-only controls (Figure 7A–C). Fluorescence was restricted to tissue pieces, appearing morphologically as a spotty pattern. Specificity controls show decreased to essentially no signal when primary antibodies were inhibited prior to incubation (a control for non-specific paratopes in the polyclonal primary antibodies; Figure 7D) and when tissue sections were digested with collagenase A for 3–6 h prior to exposure to primary antibodies (a control for non-specific binding of the primary antibodies to molecules other than the target protein; Figure 7E). As also found by Schroeter [59] and Ullmann et al. [20], and references therein), digestion with collagenase A for 1 h initially increased signal slightly.

Unexpectedly, modern *Alligator* bone demineralized with HCl exhibited relatively dimmer fluorescence than previous *Alligator* samples demineralized with EDTA and treated with the same primary antibodies (Figure 7F–H; cf. [59], Figure 4D). The expected binding pattern, showing visible bone tissue structures, (e.g., Haversian systems), also appeared patchier than has been observed in previous studies using this antibody (cf. [59], Figure 4D). This patchy binding obliterated most Haversian systems and other recognizable histologic features, leaving instead an irregular fluorescence pattern (Figure 7G,H) somewhat similar to that observed in the fossil *Thoracosaurus* bone (Figure 7B,C). However, despite these apparent artifacts of protocol-induced degradation (see Discussion), the modern *Alligator* tissue sections still exhibited stronger fluorescence than did the fossil samples. Inhibition and digestion controls for the modern samples (see Appendix A), also each dramatically diminished the signal (including after only 1 h of digestion; Figure 7I,J), confirming the specificity of antibody binding in our assays.

## 4. Discussion

Our findings cumulatively support the recovery of endogenous soft tissues and biomolecules from 63–66-million-year-old vertebrate fossils from EFP. As osteocytes were found only in fossil samples and still embedded within bony tissues, it is parsimonious to infer that these structures originated from the fossil material and not the environment. Additionally, multiple assays found only fossil samples to yield organics that reacted positively with antibodies raised against *Alligator* and chicken collagen I. Therefore, the unique depositional environment recorded by sediments of the Hornerstown Formation is the latest in the expanding list of paleoenvironmental settings shown to allow molecular preservation over geologic time. 

Wiemann et al. [10] previously demineralized a vertebra from an indeterminate crocodilian (YPM 656) that was collected from the Navesink Formation, which underlies the Hornerstown Formation. In stark contrast to our results, demineralization of YPM 656 failed to yield any cells or soft tissues [10]. There are several possible reasons for this difference. No vertebrae were analyzed in the present study, but even the cancellous turtle shell bones examined herein yielded demineralization products (Table 2), implying that it is unlikely that factors related to skeletal element type alone could account for the lack of recovery of soft tissues from YPM 656 [10]. Although the Navesink Formation also contains iron-rich glauconite, it is in a lower percentage than the Hornerstown Formation [30]. It is thus possible that sediments of the Navesink Formation did not form as conducive of a diagenetic environment for molecular preservation as those of the Hornerstown Formation (cf., [33]). It is also possible that, as observed in the disparity of microstructural preservation between light and dark fossils recovered from the Hornerstown Formation, preservation in the Navesink may be variable as well. Histological examination of YPM 656 may show that its microstructure was poorly preserved despite its intact gross morphology, similar to the light tan fossils of the Hornerstown Formation that also often failed to yield demineralization products herein (Table 2). Additionally, Wiemann et al. [10] acquired their sample from collections, meaning it was not collected fresh for the purposes of paleomolecular analyses. Though historical specimens in collections have yielded results, (e.g., [6]), it has been previously suggested that fresh samples yield better soft tissue and biomolecular recovery ([60] and references therein). However, at this time, it is not possible to concretely resolve which of these processes are responsible for the lack of recovery reported by Wiemann et al. [10].

At EFP, the majority of the Hornerstown Formation currently lies below the natural water table, (e.g., the MFL is positioned ~20 ft beneath it), and studies of regional sequence stratigraphy [61] imply that fossils within the Hornerstown Formation at this locality have likely spent tens of millions of years under saturated conditions. Despite this, we successfully recovered cells and soft tissues from numerous MFL fossils and collagen I from RU-EFP-00006-11. Thus, our findings corroborate other recent studies [3,6,10,16,35] which refute the traditional hypothesis that marine paleoenvironments are inconducive to biomolecular preservation due to hydrolysis caused by constant exposure to water [25,26]. As iron is hypothesized to aid in molecular preservation in many cases [3,32,33], it is possible that the high concentration of glauconite, a mineral rich in iron, aided in the preservation of these endogenous organics, (e.g., via iron free radical-induced molecular crosslinking [33]). It is also possible that the presence of abundant dissolved iron was responsible for some of the analytical challenges encountered herein (see below). Other authors [62] have alternatively suggested that iron may precipitate around cells and other soft tissue microstructures preserved via other pathways during early diagenesis, (e.g., alumino-silicification), forming a mineralized coating later in diagenesis. At this time, there is no evidence that these preservation pathways are mutually exclusive; indeed, at some localities, (e.g., [62]) both may contribute to long-term preservation at the molecular level.

### 4.1. Overall Preservation

Our histological analyses found light bone to be heavily degraded, having largely lost its original microstructure (HI = 0–2). This histologic alteration appears responsible for light bones yielding few cells and soft tissues upon demineralization (Table 2). For example, the only specimen to not yield any soft tissue products (crocodile tibia RU-EFP-00030-1) is entirely light in cross-section. The other specimen with this condition (*Taphrosphys* peripheral RU-EFP-02175) only yielded rare osteocytes, and crocodile humerus RU-EFP-00030-2 also only yielded rare osteocytes likely because it appears to be preserved in a state of partial degradation. Additionally, when present, the osteocytes recovered from light bones were typically poorly preserved, (i.e., with only short, stubby filopodia) compared to those from dark bones retaining well-preserved histology (HI = 4–5), which more frequently retain long, branching filopodia with multiple ramifications. The only other specimen to yield rare osteocytes (and no vessels or fibrous matrix) was turtle peripheral RU-EFP-02245. Although this specimen only exhibits a thin external rim of light bone, it is primarily composed of highly porous cancellous bone; its accordingly low volume of bone material may have reduced the chances of recovering soft tissues from this specimen, regardless of its level of preservation. All other specimens with a thin layer of light bone or composed entirely of dark bone yielded a consistently greater recovery of cellular and soft-tissue microstructures (Table 2). 

Based on our results, there appears to be no association (for these 12 specimens) between soft tissue preservation and either taxon, skeletal element, quality of macroscopic preservation, cortical thickness, or bone tissue type, (i.e., cortical vs. cancellous bone). Of the limb bones sampled, only two yielded abundant cellular and soft tissue microstructures even though all are well-preserved at the gross-morphology level. At this locality, and based on current information, the best predictor of histologic and soft tissue preservation appears to be bone color: bone tissues that are dark are histologically well-preserved bones and generally yield far greater soft tissue recovery upon demineralization. A correlation between morphologic quality and molecular preservation has been suggested previously [25,38,39,63,64], and our results further support this hypothesis. However, it remains unclear what is causing differential degradation among bones preserved within the same probable mass death assemblage [28,30] in the same horizon of the same geologic stratum. It is possible that the light degradation is a late-diagenetic artifact resulting from the modern acidic groundwater of southern New Jersey. Alternatively, these regions of poorly preserved histology within bone could result from microbial activity, as suggested by the presence in some specimens of potential MFDs and Wedl tunnels extending from the light outer layer into the underlying, better-preserved portions of dark bone (cf., [65,66,67]). This taphonomic question is currently under investigation as part of a separate study. 

### 4.2. Soft Tissue Preservation

Demineralization in EDTA in this study proceeded more slowly and generally yielded fewer microstructures than from other similarly aged fossil bones we have tested [11,59]. Slow demineralization in EDTA was also found by Norris et al. [68] for a Permian *Dimetrodon* bone. As in our study, these authors were able to successfully demineralize their specimen in a solution containing HCl. The cause of such cases of slow demineralization in EDTA remains unknown. Powder XRD analyses identified RU-EFP-00006-11 as being primarily composed of fluorapatite (Figure 3A), indicating that micro-scale permineralization by secondary mineral phases containing non-divalent cations, (e.g., iron oxides with Fe^3+^ ions or quartz with Si^4+^ ions) which cannot be chelated by EDTA is an unlikely explanation. It is possible that significant substitution of trivalent cations for divalent Ca^2+^ ions in bone apatite could hinder the demineralization process (as EDTA only chelates divalent cations; [50]), but further analyses, (e.g., trace element analyses) would be required to evaluate this potential cause. HCl provides harsher, more acidic conditions for demineralization; therefore, demineralization trials employing this solution required significantly less time and yielded a greater recovery of soft tissues. However, as HCl incubation is also a step in our protein extraction protocol (see Appendix A), incubation of fossil bone fragments in this acid may result in solubilization of proteins and other biomolecules which lack divalent cations. Although microstructures isolated by demineralization with HCl did not appear under transmitted light to visually differ from those recovered using EDTA, the effects of HCL versus EDTA demineralization warrants further investigation. 

Osteocytes were the most abundant microstructures recovered from our EFP specimens. The common lack of recovery of microstructure morphologically consistent with blood vessels, at least in our crocodilian samples, may be due to the rarity of vessels in the original cortical/cancellous bone tissue. The cortex in modern crocodilians is not as vascular as that of non-avian dinosaurs [69,70,71] or modern birds; as a result, fossil crocodile bone would not be expected to yield as many structures morphologically consistent with vessels as would dinosaur bone. It should also be noted that although the external cortex of femur RU-EFP-00006-11 has been perforated by occasional Wedl tunnels, the vessel structures we recovered do not appear to represent biofilm coatings of these tunnels because they retained their structure after demineralization, and each possess a clear lumen [72]. Low recovery of the fibrous matrix may relate to the depositional environment, but more testing would be required to elucidate such a causal connection. 

### 4.3. Biomolecular Preservation

Taken together, our molecular assays support the conclusion that the soft tissues and collagen I recovered from this specimen are endogenous. Though a PAGE with silver-stain assay is not a specific test for collagen I or proteins, it can identify the presence and molecular weights of organic compounds in a sample, making it an informative screening assay for paleomolecular studies [4,20,73]. Both fossil AMBIC extract lanes showed a distinct increase in coloration after silver-staining, whereas all negative control lanes (sediment, extraction blank, and Laemmli buffer) exhibited little to no coloration (Figure 5A,B). However, the prestaining of the gel likely points to diagenetic humic substances being present in this sample [74]. Humics can bind silver [75,76], which may have caused the increase in staining. Though this pattern of silver nitrate binding cannot directly support the presence of protein in the sample, this pattern does not falsify the presence of protein in this fossil (as no staining would). Thus, we continued to analyze this sample with assays of greater specificity that are not susceptible to humics. As predicted, the modern *Alligator* extracts exhibited less smearing and better banding than the fossil extracts (Figure 5C; see Appendix A for further discussion of our silver-staining results).

Given the recognition of unique, albeit non-specific, organics in RU-EFP-00006-11 by silver-staining, we next performed ELISA and in situ immunofluorescence as independent assays to identify if the primary structural protein collagen I was present in both whole-bone extracts and demineralization products, respectively. These assays complement one another as ELISA is approximately an order of magnitude more sensitive than immunofluorescence [73,77], whereas immunofluorescence allows in situ localization of epitopes in native tissues [15,73]. Fossil extracts in ELISA assays yielded well over double the absorbance of the sediment and extraction buffer controls (Figure 6), supporting the presence of endogenous collagen I in RU-EFP-00006-11. Negative absorbance values for sediment and extraction controls indicate these samples were less reactive than the group blank (PBS only), demonstrating the collagen signal in bone wells is not attributable to these sources of possible contamination. Additionally, the absorbance values of all samples incubated with only secondary antibodies were negligible, (i.e., significantly less than twice the absorbance of the fossil samples incubated with both primary and secondary antibodies), removing non-specific secondary antibody binding (such as to humic substances) as a cause for the high positive signal for the fossil bone extracts. Thus, this assay supports the presence of endogenous collagen I in the femur of this *Thoracosaurus*. 

Modern *Alligator* tissue samples analyzed by in situ immunofluorescence exhibited fluorescence when exposed to polyclonal antibodies raised against chicken collagen I (Figure 7G,H). Together with the reduced signals observed in our specificity controls (Figure 7I,J; Appendix A), these results agree with those of Schroeter [59] who concluded that the collagen I epitopes in these two extant archosaurs are conserved enough in structure to each be recognized by the primary antibodies used herein. Because these antibodies can successfully detect collagen I in modern *Alligator*, they were predicted to also bind to fossil crocodilian collagen I. The fluorescence signal exhibited by the fossil tissues was lower in intensity compared to the modern *Alligator* (Figure 7B,C), yet still far brighter than in secondary-only controls (Figure 7A). Fluorescence in fossil tissue sections decreased in all specificity controls (Figure 7D,E; Appendix A), again supporting the specific binding of the primary antibodies to epitopes of collagen I. Both modern and fossil tissues exhibited patchy binding patterns, possibly due to tissue degradation from demineralization with HCl or its ability to solubilize proteins (as in our extraction protocol), as well as decay inherent in fossilization for RU-EFP-00006-11.

## 5. Conclusions

Our paleomolecular investigations of fossils preserved in the glauconitic, shallow-marine depositional environment of the Hornerstown Formation at EFP add a unique paleoenvironment to the growing list of those which have yielded biomolecular and soft tissue preservation within fossil bones. This depositional environment was rich in iron at the time of burial and remained rich in iron due to the glauconite-dominated composition of the sediments and influx of dissolved iron from recent groundwaters. We hypothesize that this abundant supply of iron may have facilitated soft tissue and biomolecular preservation in these specimens via the free radical-induced crosslinking reactions elucidated by Boatman et al. [33]. Soft tissues were recovered by demineralization with HCL and EDTA, though HCL treatments took less time and were required for some samples. Though all specimens we examined are well-preserved in terms of gross morphology, multiple exhibit varying amounts of alteration in the form of light tan-colored bone which was found to be histologically degraded and thus yielded minimal cells and soft tissues upon demineralization. Association of poor histological preservation with poor soft tissue recovery is logical, and it appears that at EFP this alteration may have occurred, at least in part, by microbial degradation. All biomolecular assays completed on RU-EFP-00006-11 support the presence of endogenous collagen I in this *Thoracosaurus* femur. PAGE with silver-staining exhibited evidence for organics in this fossil, and ELISA and in situ immunofluorescence results each independently support retention of collagen I in this specimen. Collectively, our results and those of previous authors [3,6,10,16,35] support the conclusions of Nielsen-Marsh and Hedges [78] and Hedges [38] that relatively constant immersion in water may not preclude endogenous molecular preservation in fossil bones. As a result, there are many additional paleoenvironments yet to be explored that may preserve endogenous biomolecules and soft tissues.

## Figures and Tables

**Figure 1 biology-11-01161-f001:**
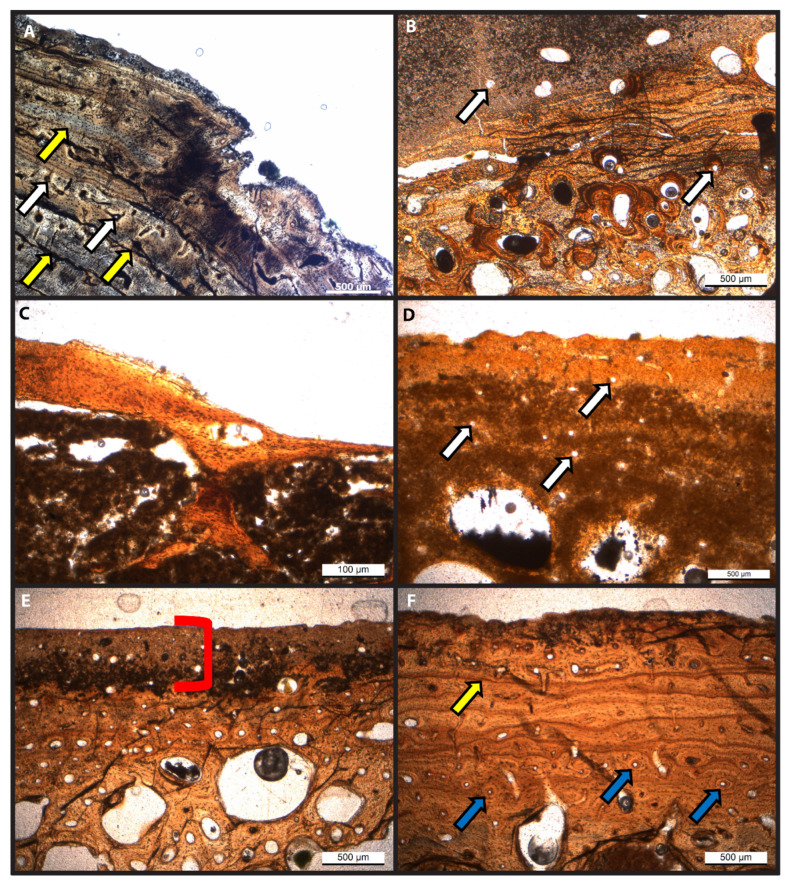
Thin sections of specimens used for demineralization illustrate the varying degrees of bone microstructure preservation. (**A**) *Thoracosaurus* femur (RU-EFP-00006-11) femur showing well-preserved bone microstructure including Sharpey’s fibers (Histologic Index [HI] = 5). (**B**) Indeterminate crocodilian tibia (RU-EFP-00030) showing little bone microstructure preserved (HI = 1). (**C**) *Euclastes* costal prong (RU-EFP-00018) showing excellent preservation (HI = 5). (**D**) *Taphrosphys* peripheral (RU-EFP-02175) showed very poorly preserved microstructure (HI in this region = 1; HI for the entire thin section = 2). (**E**) Indet. Pan-Cheloniid peripheral (RU-EFP-02245) showed well preserved internal cortex but a highly altered outer cortex (HI = 4). (**F**) Juvenile *Taphrosphys* costal (RU-EFP-02222) with well-preserved microstructure (HI of this region and the entire thin section = 4–5). White arrows indicate vascular canals, yellow arrows indicate lines of arrested growth, and blue arrows indicate primary osteons. The red bracket corresponds to external rims of light color and poor histological integrity.

**Figure 2 biology-11-01161-f002:**
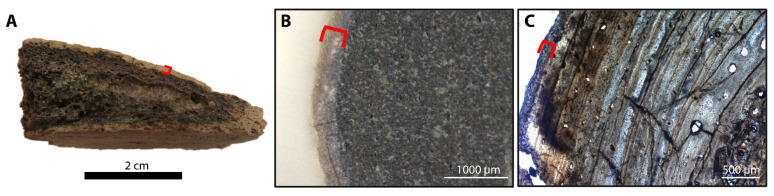
Examples of the two bone colors observed in specimens from EFP: an outer rim of light-colored bone (indicated by red brackets) and a darker interior. (**A**) Hand sample of a bone fragment from EFP. Note the light-coloration of the lower surface of the bone (directly above the scale bar). (**B**,**C**) *Thoracosaurus neocesariensis* femur (RU-EFP-00006-11) in (**B**) thick section and (**C**) thin section. Images (**B**,**C**) are not from the same position along the circumference of this bone, and the light-colored rim is not a consistent thickness along the entire circumference. Scale bars as indicated.

**Figure 3 biology-11-01161-f003:**
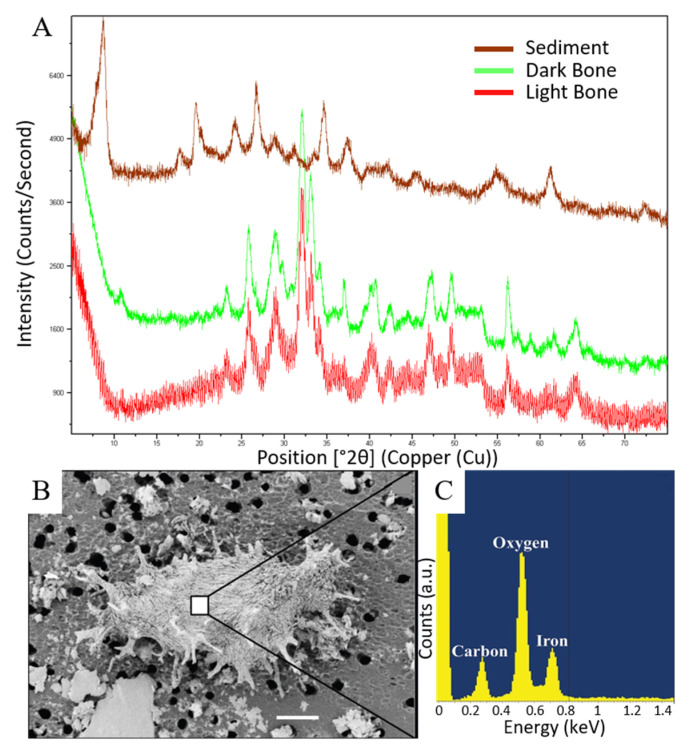
Mineralogical analysis of sediment, dark bone, and light bone samples from EFP and SEM-EDS of a structure morphologically consistent with an osteocyte from RU-EFP-00006-11. (**A**) Diffractogram of XRD results showing distinct differences between the sediment and fossils, which were identified as glauconite and fluorapatite, respectively; (**B**) SEM micrograph of a structure morphologically consistent with an osteocyte; (**C**) EDS spectrum from center of osteocyte (square area) in (**B**). The large peak surrounding 0 keV is an artifact of the very low count rates from the osteocyte. Scale bar 4 µm in (**B**).

**Figure 4 biology-11-01161-f004:**
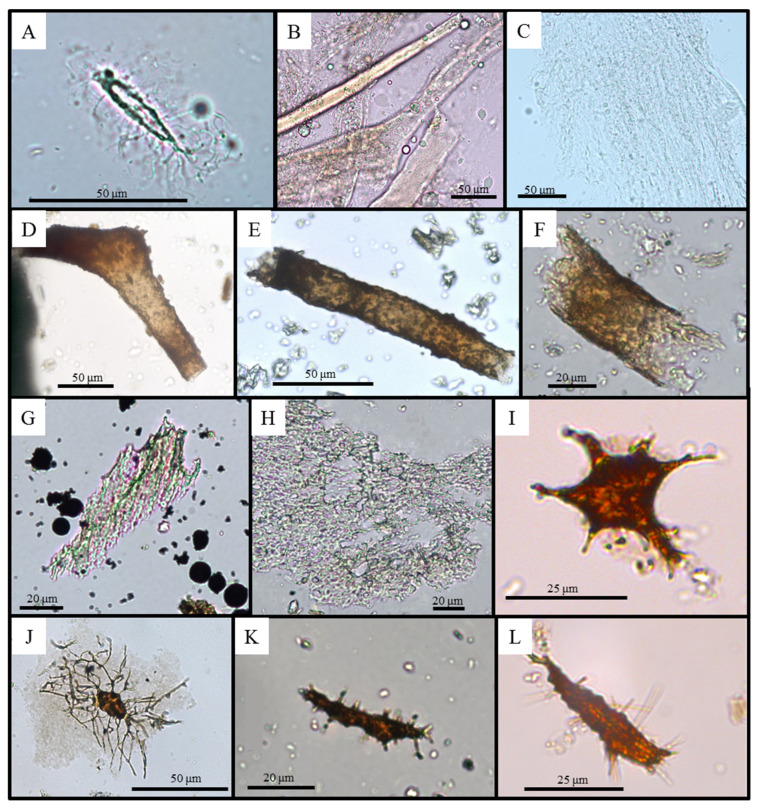
Modern and fossil demineralization products. (**A**) Modern American alligator osteocyte. (**B**) Alligator blood vessels. (**C**) Alligator collagen matrix. (**D**,**E**) *Thoracosaurus* (RU-EFP-00006-11) blood vessels. (**F**) *Taphrosphys* (RU-EFP-00002-2) blood vessel. (**G**) *Thoracosaurus* (RU-EFP-00006-8) collagen matrix. (**H**) *Taphrosphys* (RU-EFP-00002-2) collagen matrix. (**I**) *Taphrosphys* (RU-EFP-02222) stellate osteocyte. (**J**) Testudines indet. (RU-EFP-04161-8) stellate osteocyte. (**K**) *Taphrosphys* (RU-EFP-04162) flattened oblate osteocyte. (**L**) *Euclastes* (RU-EFP-00018) flattened oblate osteocyte.

**Figure 5 biology-11-01161-f005:**
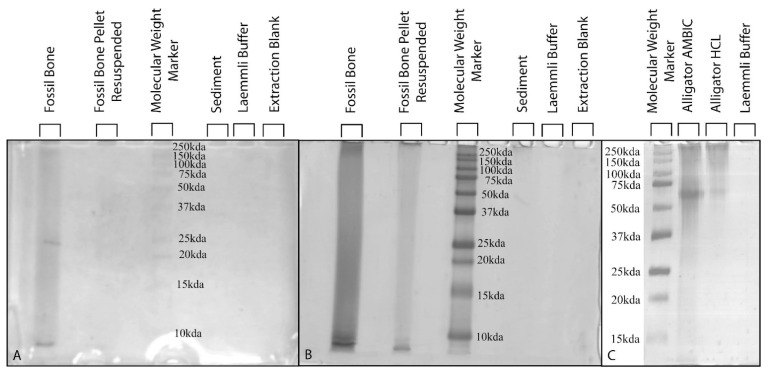
Polyacrylamide gel electrophoresis (PAGE) with silver-staining of fossil and modern samples. (**A**) PAGE of fossil samples prior to silver-staining, exhibiting pre-staining in the fossil bone lane. Fossil samples were loaded at 50mg of pre-extracted bone per lane; (**B**) same gel as (**A**) after development with silver nitrate. Both fossil sample lanes exhibit increased staining, whereas no binding is visible in sediment, Laemmli buffer, and extraction blank lanes; (**C**) silver-staining of modern *Alligator* samples, loaded at 20 µg/lane.

**Figure 6 biology-11-01161-f006:**
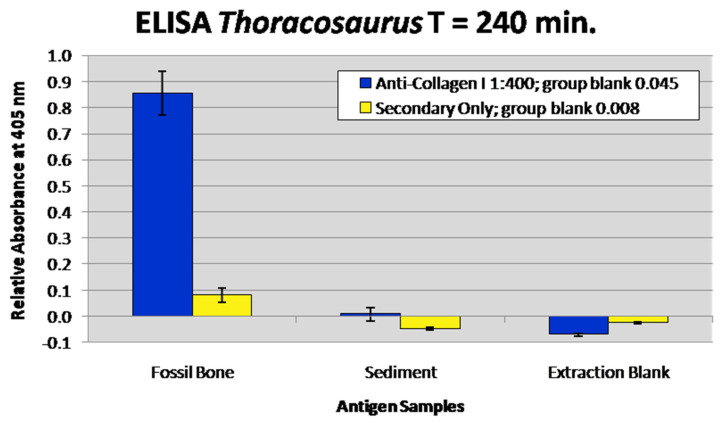
Enzyme-linked immunosorbent assay results of fossil and sediment chemical extracts. Extraction buffers were also run as a negative control to test for contaminants. Fossil and sediment samples were loaded at 200 mg of pre-extracted weight. Blue columns represent absorbance at 405 nm at 240 min, with incubation in anti-chicken collagen I antibodies at a concentration of 1:400. Yellow columns represent the absorbance of the secondary-only controls for each sample. The fossil bone sample is over three times its secondary control and the sediment and extraction blank show low absorbance, indicating that they are not a source for the positive signal in the fossil sample. Error bars represent one standard deviation of the mean absorbance of each sample.

**Figure 7 biology-11-01161-f007:**
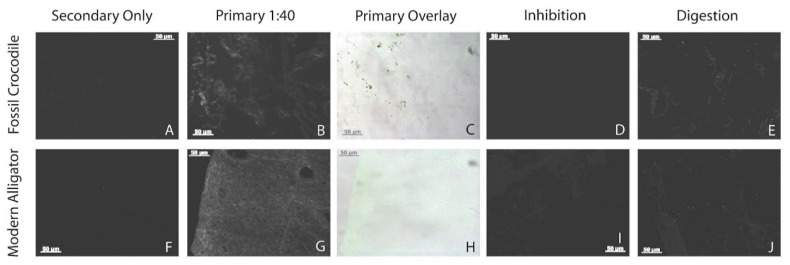
In situ immunofluorescence results for fossil and modern *Alligator* samples. All sections imaged at 200 ms exposure. (**A**,**F**) Secondary only negative-control tissue sections never exposed to primary antibodies; (**B**,**G**) tissue sections incubated with anti-chicken collagen I antibodies at 1:40 concentration; (**C**,**H**) overlays of fluorescence images in (**B**,**G**) on light microscope images, showing localization of fluorescence to tissue within sections (fluorescence shown as green coloration); (**D**,**I**) tissue sections treated with the same anti-chicken antibodies but exposed to *Alligator* collagen prior to incubation with sections (“inhibition control”); (**E**,**J**) tissues sections incubated with collagenase for 3 h prior to incubations with the same primary antibodies. Negative controls and specificity controls exhibit reduced or absent signals, indicating a lack of spurious binding and the presence of endogenous collagen I in each bone sample. Scale bars as indicated.

**Table 2 biology-11-01161-t002:** An overview of the 12 samples analyzed in this study. Abbreviation: Fm = Formation.

Specimen RU-EFP-#	Taxon	Element	Source Horizon/Fm	Osteocyte Recovery	Vessel Recovery	Fibrous Matrix Recovery	Color	Histology Assessed
00002-2	*Taphrosphys*	Peripheral	MFL	Abundant	Frequent	Frequent	Thin Light Rim	Yes
00006-8	*Thoracosaurus*	Scute	MFL	Frequent	Absent	Rare	Thin Light Rim	Yes
00006-11	*Thoracosaurus*	Femur	MFL	Abundant	Rare	Uncommon	Thin Light Rim	Yes
00018	*Euclastes*	Costal	MFL	Abundant	Absent	Absent	Dark	Yes
00030-1	Crocodylia Indet.	Tibia	MFL	Absent	Absent	Absent	Light	Yes
00030-2	Crocodylia Indet.	Humerus	MFL	Rare	Absent	Absent	Medium Color	No
02175	*Taphrosphys*	Peripheral	MFL	Rare	Absent	Absent	Light	Yes
02222	*Taphrosphys*	Costal	MFL	Abundant	Absent	Absent	Dark	Yes
02245	Pan- Cheloniidae Indet.	Peripheral	MFL	Rare	Absent	Absent	Thin Light Rim	Yes
02295	Testudines Indet.	Femur	MFL	Abundant	Absent	Absent	Dark	No
04161-8	Testudines Indet.	Zeugopodial	Hornerstown	Abundant	Rare	Uncommon	Dark	No
04162	*Taphrosphys*	Costal	Hornerstown	Abundant	Absent	Absent	Dark	No

# represents the generic place holder for the numbers above.

## Data Availability

All data generated by this study are available in this manuscript and the accompanying Appendix A.

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
