# Peer review of "Soft Tissue and Biomolecular Preservation in Vertebrate Fossils from Glauconitic, Shallow Marine Sediments of the Hornerstown Formation, Edelman Fossil Park, New Jersey"

_biology, 2022, doi:10.3390/biology11081161_

Round 1

Reviewer 1 Report

Dear authors, I feel like your article has scientific merit, however it has a number of issues that should be fixed prior to publication in my opinion.

My major issues are below:

- The introduction does not really explain well why it is important to study molecular preservation in marine environments. Why would we expect that soft tissues and molecules couldn't preserve in this type of environment? As far as I am concerned, nothing to me would suggest that sea water inhibits preservation. You talk about it very briefly (although NaCl is not mentioned, just 'water') in the discussion, but I feel like it should be put in the introduction. Otherwise the importance of this study would be lessened.

- Results sections: In the histological description, all the images of Figure 1 are way too small and blurry. This figure should be re-made. Moreover, there are throughout this section many descriptions without figures, so the results are not supported by any data. I suggest to add new figures, or new images to the existing figures for each description in the result section (see directly the pdf with comments).

- Results section: I cannot see the Wedl tunnels, nor MFDs. It is necessary to either show them with high-magnification histological figures, or SEM images. The SEM is the best way to identify MFDs. If you have easy access to an SEM, I strongly suggest to go take new pictures to support your identifications and increase the quality of this paper.

- Overall, it seems to me that this study does not provide any new breakthrough discovery. I also see problems with the immunohistochemistry results of both the extant and extinct crocodile (which has to do with the protocol, as you said, this is intriguing, but I understand that protocol issues do happen sometimes). The most interesting result of this study is the correlation between bone color and soft-tissue preservation in my opinion, however there is no figure to show the two different categories of bone color. I suggest to add a figure showing the gross morphology for the bones analyzed and their color.

- Throughout the manuscript, quotation marks are put next to the words 'blood vessels', 'cells' etc... these quotation marks add nothing to the paper. Actually, they even make it seem like you are unsure what these structures really are. As if perhaps these fossil 'cells' may be something else. I strongly encourage you to remove all these quotation marks, and simply call them what they are: fossilized blood vessels, or fossilized cells. They cannot be anything else (unless you could provide alternative explanation for what they are?).

- Discussion: I suggest there should be more discussion about marine vs. other types of environments. Otherwise this paper is simply a description of soft tissues and molecules in a new specimen from a new formation but it doesn't provide any new solid scientific insight about molecular and soft tissue preservation. It is missing something, please add any new insight that you may have.  

Nevertheless, after all of this criticism, I do think you did a lot of work and this paper definitely merits to be published.

I have many additional minor comments in the pdf. I hope that they can all be addressed.

Best wishes and good luck for the revisions of this interesting paper that I am sure will become excellent after some modifications.

Author Response

We have addressed the reviewers comments. Their comments are copied and italicized below, followed by our responses.

My major issues are below:

- The introduction does not really explain well why it is important to study molecular preservation in marine environments. Why would we expect that soft tissues and molecules couldn't preserve in this type of environment? As far as I am concerned, nothing to me would suggest that sea water inhibits preservation. You talk about it very briefly (although NaCl is not mentioned, just 'water') in the discussion, but I feel like it should be put in the introduction. Otherwise the importance of this study would be lessened.

We agree with the reviewer, and have therefore added to the Introduction additional background information about why molecular paleontologists historically rarely studied marine fossils, as this helps explain the importance of our results (recovering endogenous collagen from a marine fossil).

Lines 56-61: “Initially only terrestrial environments were investigated, as it was thought that hydrolysis from continuous exposure to water would be inconducive to molecular preservation [25,26]. More recently, a few studies examined aquatic taxa or fossil specimens preserved in marine environments (e.g., [6,16,27]). However, the number of geologic formations and types of host lithologies, and, therefore, the types of depositional paleoenvironments investigated, still each remain limited.”

- Results sections: In the histological description, all the images of Figure 1 are way too small and blurry. This figure should be re-made. Moreover, there are throughout this section many descriptions without figures, so the results are not supported by any data. I suggest to add new figures, or new images to the existing figures for each description in the result section (see directly the pdf with comments).

We have taken the reviewers suggestions to improve Figure 1 and have added 3 new Supplementary figures to address the data not originally visualized, as recommended in the PDF comments.

- Results section: I cannot see the Wedl tunnels, nor MFDs. It is necessary to either show them with high-magnification histological figures, or SEM images. The SEM is the best way to identify MFDs. If you have easy access to an SEM, I strongly suggest to go take new pictures to support your identifications and increase the quality of this paper.

Unfortunately, an SEM is not available during the short time frame allotted for revision to collect such images, but we have added high magnification histologic images (commonly used to show such structures) from light microscopy in Figure S2 that clearly show the structures we discuss.

- Overall, it seems to me that this study does not provide any new breakthrough discovery. I also see problems with the immunohistochemistry results of both the extant and extinct crocodile (which has to do with the protocol, as you said, this is intriguing, but I understand that protocol issues do happen sometimes). The most interesting result of this study is the correlation between bone color and soft-tissue preservation in my opinion, however there is no figure to show the two different categories of bone color. I suggest to add a figure showing the gross morphology for the bones analyzed and their color.

First, as stated in our Abstract and Conclusion, our findings corroborate a growing list of studies which demonstrate that fossil bones preserved in marine environments can yield endogenous soft tissues and biomolecules, and our study also documents molecular preservation in a unique environment. Both of these factors suggest that molecular paleontologists should continue to test fossils from new depositional environments because the limits on preservation are yet to be discovered. Whether this can be considered as a “breakthrough discovery” is subjective and irrelevant, as the reviewer also states elsewhere that they feel our manuscript has scientific merit. Regarding our immunohistochemical results, we have addressed this as much as possible at this time, and it is an interesting finding which we hope further studies will continue to shed light on. Finally, we have accepted the reviewer’s suggestion to add a new image of the gross morphology of a two-color bone (in hand sample) as a panel in Figure S1.

- Throughout the manuscript, quotation marks are put next to the words 'blood vessels', 'cells' etc... these quotation marks add nothing to the paper. Actually, they even make it seem like you are unsure what these structures really are. As if perhaps these fossil 'cells' may be something else. I strongly encourage you to remove all these quotation marks, and simply call them what they are: fossilized blood vessels, or fossilized cells. They cannot be anything else (unless you could provide alternative explanation for what they are?).

We are glad the reviewer agrees that the field can move beyond this outdated standard; however, we have not formally tested the osteocytes and vessels to confirm that they are comprised of original cells/tissue as opposed to mineralized casts. Thus, we have removed all quotations as the reviewer suggested and have added the text below to clarify:

Line 329-333 “Although further molecular testing is needed to confirm that these structures represent original cells/tissues, we hereafter refer to them as osteocytes and vessels for brevity. Our antibody assays identified the presence of endogenous collagen I in pieces of fibrous matrix (see below), supporting the endogenous nature of these tissue fragments.”

- Discussion: I suggest there should be more discussion about marine vs. other types of environments. Otherwise this paper is simply a description of soft tissues and molecules in a new specimen from a new formation but it doesn't provide any new solid scientific insight about molecular and soft tissue preservation. It is missing something, please add any new insight that you may have.  

As reviewed in our Introduction section of the text, we are not the first to recover signatures of endogenous biomolecules in fossil bones from marine sediments of Mesozoic age (prior reports include Lindgren et al. [2011] and Surmik et al. [2016]), but we are the first to recover biomolecular data from a fossil bone preserved in unconsolidated glauconitic greensands. In the Discussion section of the manuscript we briefly discuss what we feel are the two most significant implications of this discovery, which are that: 1) an abundance of iron appears to have facilitated/aided molecular preservation within the specimen we analyzed, which bolsters the hypotheses of Schweitzer et al. (2014) and Boatman et al. (2019) that iron free radical-induced crosslinking promotes biomolecular stabilization, and; 2) our discovery corroborates the conclusions of Lindgren et al. (2011) and Surmik et al. (2016) that protracted submersion in seawater, marine pore fluids, and phreatic groundwater does not necessarily preclude biomolecular preservation, even in bones of Mesozoic age. In the absence of any more specific recommendations from the reviewer, we feel that the Discussion text already concisely discusses the above points, so we have made no changes nor additions to the Discussion section. Additionally, our manuscript’s emphasis is not on molecular taphonomy or diagenesis, so we therefore only briefly discuss these topics.

Nevertheless, after all of this criticism, I do think you did a lot of work and this paper definitely merits to be published.

I have many additional minor comments in the pdf. I hope that they can all be addressed.

Thank you and we have addressed these in the manuscript.

Reviewer 2 Report

In this study,  Voegele and colleague investigated the preservation of soft tissue and its molecules in vertebrate fossils. Because there are many serious issues in biochemical analyses, it is unconvinced their conclusions.

1.English editing is necessary.

2. Methods part is too simple. In particular, 2.2.6~2.2.8 are egregious

3. 1. Fig.5; There are too few bands. Therefore, it is strongly suspect from this figure that proteins are not preserved in the fossils.

4. Fig.6; It is insufficient from these data to conclude the preservation of Collagen I in  the fossil. It is stronlgy suspected that degraded materials cross-reacted to the antibody. To demonstrate that, the authors must perform immunoblot analyses.

5. Fig. 7; Quality of immuohistochemistry is too low. It is difficult to discriminate the positive signal.

Author Response

We have addressed the reviewers five comments below. Their words are coppied from their review and are in italics below.

1.English editing is necessary.

As all other reviewers rated our manuscript as “English language and style are fine/minor spell check required” and Reviewer 2 provides no other details, we feel that by correcting the few instances indicated by the other reviewers that needed increased clarity we have sufficiently addresses this comment.

2. Methods part is too simple. In particular, 2.2.6~2.2.8 are egregious

Our Methods section represents almost 1/5th of our main text length and half of our Supplemental Material for a total of roughly 8 pages of text. As none of our methods are new, we cite 7 papers that contain detailed outlines of these protocols. This is convention and respects not using space in journals to publish what has already been published, while giving credit to those who developed the protocol. We feel our methods are more than sufficient.

3. 1. Fig.5; There are too few bands. Therefore, it is strongly suspect from this figure that proteins are not preserved in the fossils.

We do not claim that this assay alone indicates protein preservation or even the presence of proteins specifically. As we acknowledge on Lines 598-601, this assay does not test for collagen or proteins specifically and we used it as a screening method for organics before proceeding to more expensive techniques. Also, as stated on lines 606-609, our result does not preclude the presence of proteins and was suggestive of more investigation – and we conducted such investigation with two additional assays.

4. Fig.6; It is insufficient from these data to conclude the preservation of Collagen I in  the fossil. It is stronlgy suspected that degraded materials cross-reacted to the antibody. To demonstrate that, the authors must perform immunoblot analyses.

We are unconvinced an immunoblot will add significant elucidation to this point as collagen sequences obtained through mass spectrometry of fossil bone protein extracts (e.g., Asara et al. 2007; Schweitzer et al. 2009; Schroeter et al. 2017) have recovered only fragments of the collagen molecule (i.e., a few diagnostic peptides). Protein extracts from these same samples have also shown antibody binding (see Schweitzer et al. 2007a, 2009). Thus, it is likely that preserved collagen recovered from the Thoracosaurus specimen herein would not be fully intact molecules either, and likely not even “large” fragments of its original peptide sequence. Indeed, as a result of (common) hydrolytic and oxidative decay, it is parsimonious to expect degraded molecular remnants in fossils (see, e.g., Cleland et al. 2015, 2016, 2021; Schroeter et al. 2017). Additionally, collagen is known to readily form crosslinks, and previous studies have suggested that early-diagenetic crosslinking could aid in molecular preservation (e.g., Schweitzer et al. 2007b, 2014; Boatman et al. 2019). Crosslinking would also alter the molecular weight of proteins and their decay products as they aggregate together, potentially allowing antibodies to bind at varied molecular weights other than that of the full, intact molecule (reminiscent of the smear in our AMBIC extract lanes in our PAGE with silverstain results). Further, our experimental controls each came back negative as would be expected if our primary antibodies were binding as they should, in other words only to archosaurian collagen I epitopes. If, as suggested by the reviewer, degraded fossil material was cross-reactive and binding with the primary antibody, it would also most likely be expected to do the same with the secondary antibody. As we’ve stated and shown, our secondary-only negative controls are negative and show no such binding. Finally, we do not use just ELISA to support the preservation of endogenous collagen in this fossil, but complement these findings with positive results from immunofluorescence.

5. Fig. 7; Quality of immunohistochemistry is too low. It is difficult to discriminate the positive signal.

We acknowledged in our manuscript that our immunofluorescence data from our extant Alligator are not as bright as would typically be expected, but that there is still clear fluorescence unambiguously indicative of a positive result. We also have given a plausible rationale to explain these immunofluorescence results and another reviewer has commented that they accept this rationale. All fossil samples yet investigated by immunofluorescence have yielded a lower signal than extant controls (e.g., Schweitzer et al. 2007, 2009; Cleland et al. 2015; Pan et al. 2016, 2019; Ullmann et al. 2020). This is logically expected as it has been shown that preserved proteins are usually only fragments (i.e., isolated, partial peptides; see references in response to previous comment above), and because antibodies are made to recognize modern taxa, not the fossil taxon under study (and are therefore not perfectly “matched” to the structure of the target protein in the fossil taxon). Despite these challenges, there is clearly a positive signal in our bone samples (Figure 7B,C,G,H) and not in our secondary-only controls (Figure 7A,F). Additionally, all of our specificity tests (Figure 7D,E,I,J) yielded dramatically-reduced signals compared to the bone sample positive controls. Cumulatively, these results constitute a positive signal for the presence of collagen I in the Thoracosaurus specimen. We have maximized the contrast of the panels in Figure 7 to show this positive signal as best as we can, and feel that it is sufficient.

Reviewer 3 Report

This manuscript is so well done in its clear writing, careful scholarship, and thorough documentation on the soft tissue and biomolecular preservation of fossils in a new type of environment previously unexplored that I do not have anything to  critique. The authors made a solid case on expanding the range of ancient environments where soft tissues and organic molecules can be preserved. I congratulate the authors for such a well-done work!

Author Response

Thank you for such high praise for our manuscript. 

Reviewer 4 Report

My comments are shown in the attached file. This is an interesting paper, well worth publishing.

Author Response

We have addressed the reviewers comments provided in their pdf. Their comments are copied and italicized below, followed by our answers.

The Title, Simple Summary, and Abstract all fail to list the geologic age of the formation that contained the fossils. Line 73 of the Introduction lists the age as Maastrichtian-Danain , without mentioning that these are late Cretaceous-early Paleocene. This is in contrast to the entries in Table 1, where ages are listed as epoch or period.

We understand the reviewer’s request for the important information of the age of the fossil to be more prominent. We have added the age to the Simple Summary and Abstract and clarifying details to the Introduction. No additions were made to the Title (as it is already very long). Additionally, no changes were made to Table 1.

Line 30-32: “We conducted paleomolecular analyses of shallow marine vertebrate fossils from the Cretaceous–Paleogene Hornerstown Formation, an 80–90% glauconitic greensand from Jean and Ric Edelman Fossil Park in Mantua Township, NJ.”

Line 75-77: “All specimens were recovered from the Late Cretaceous (Maastrichtian)–early Paleogene (Danian) Hornerstown Formation.”

 Line 15 describes samples being “submerged in acid” to achieve deminieralizion. However, the later methods section describes the use of EDTA for the initial demineralization EDTA is weak acid, demineralization results from chelation, not acid attack. Most samples were immersed in HCL as a second step, so a more correct description would be to say that demineralization was accomplished using a combination of chelation and acid dissolution.

We understand the reviewer’s recommendation for clarity and have made changes to address this in the Simple Summary. However, we are concerned that chelation is too technical for a Simple Summary and have changed “submerged” to “demineralized” to more accurately reflect our methods while avoiding jargon.

Line 16-17: “Twelve samples were demineralized in acid to remove the mineral component of bone.”

To add clarity we’ve also modified the text in the Methods section 2.2.3 to make this distinction.

Line 191-192: “A roughly 0.5 cm3 fragment of each fossil was initially submerged in 0.5 M disodium ethylenediaminetetraacetic acid (EDTA) to chelate calcium.”

Line 197-199: “After 1–2 months in EDTA, if demineralization was still mostly ineffective, then as necessary samples were demineralized by acid dissolution in 0.2 M or 0.5 M hydrochloric acid (HCl) for several days to two weeks, …”

Figure 1. The caption reports HI values, but the term is not defined until later in the text (lines 175, 176). The “overall bone” value is not defined.

We have defined HI in the figure caption and replaced “overall bone” with “the entire thin section” for clarity.

Figure 3. The caption merely says that the XRD patters are distinctly different between sediment and fossils. The text explains that the sediment is composed of glauconite, the fossil bone consists of fluorapatite. These identifications need to be included in the caption. A minor point: the vertical axis is labeled as “intensity counts”, the intensity numbers shown for axis labeling are counts/second. The horizontal scale is shown in two-theta angles rather than d-spacings in angstrom units. The problem with using two-theta values is that they are different depending on the target material used in the X-ray tube. These values are from a Cu target tube filtered to produce K-alpha radiation. This information needs to be included in the methods section and/or the figure caption.

We have modified the axes of Figure 3A and caption as suggested by the reviewer. The Cu tube and K-alpha radiation information is already in our XRD methods section see lines 185-187 copied below:

“Each powdered sample was then loaded into a sample holder and analyzed using Cu Kα radiation (λ = 1.54178AÌŠ) and operating at 45 kV and 40 mA.”

Figure 3C. EDS spectrum The axes are not labeled, but the horizontal axis is clearly in KeV, the vertical in relative intensity. The low end of the spectrum has a gigantic noise peak. This kind of noise can result from a detector defect, (e.g., ice crystals trapped within a cryogenic-cooled detector). But the authors used an Oxford silica-drift detector. I use the same model, and the big noise peak occurs when specimens yield very low count rates. I think in the interests of accurate description the caption should note that this peak is an artifact. I’m also a bit puzzled why the Fe peak shown is the relatively weak L-alpha peak at 0.705 KeV, there is presumable a much higher K-alpha peak at 6.4 KeV. I am guessing that the researchers may have been using low beam voltage in the SEM to try to avoid thermal damage to the specimen. It would be good to list the KeV in the SEM methods section.

We have modified Figure 3C axes as requested by the reviewer. We have also specified that the first peak is an artifact and added the details of the SEM methods as suggested by the reviewer.

Line 211-213: “The resulting uncoated sample was imaged using a field emission-scanning electron microscope (FE-SEM Zeiss Supra 50VP) at Drexel University, operating at a working distance of 6.6 mm and at an accelerating voltage of 1 kV.”

Line 352: “The large peak surrounding 0 keV is an artifact from the very low count rates from the osteocyte.”

Line 557. osteocytes’ has an apostrophe that needs to be deleted

Accepted

Here is a recent paper that would be worth including in the references. The authors provide a good overview on the subject of preservation of organic matter in fossil bones. This chapter includes a length bibliography, including many not liste in the current manuscript. Wiersma, K.; Läbe, S.; Sander, P.M. Organic phase preservation in fossil dinosaur and other tetrapod bone from deep time. In Fossilization, Gee, C.T.; McCoy, V.E.; Sander, P.M., editors. John Hopkins University Press, Baltimore, Maryland USA, 2021, Chapter 2, pp. 16-54.

Accepted

Round 2

Reviewer 2 Report

Revised version is much improved.